# High caseload of Scabies amongst Rohingya refugees in Cox's Bazar, Bangladesh: A retrospective analysis of the epidemiological and clinical characteristics of cases, July 2022 to November 2023

**Bahaa Aldin Alhaffar**[1], **Soriful Islam**[2], **Mohammad Injamul Hoq**[3], **Asish Das**[2], **Shakil Mohammed Shibloo**[2], **Mahmudul Hasan**[2], **Kamar Uddin Muzakkir**[2], **Kawser Mahmud**[2], **Trine Grondahl Petersen**[2], **Karsten Noko**[2], **Pankaj Paul**[2], **Srijeeth S. Nair**[2], **Yves Wailly**[4], **Marta Miazek**[4], **Letizia di Stefano**[4], **Michel van Herp**[4], **Lekha Rathod**[1,4], **Amrish Baidjoe**[1,4], **Temmy Sunyoto**[1,4]*

**1** Luxembourg Operational Research and Epidemiology Support Unit, Médecins Sans Frontières, Luxembourg City, Luxembourg, **2** Médecins Sans Frontières – Jamtoli Project, Cox's Bazar, Bangladesh, **3** University of Creative Technology Chittagong, Chittagong, Bangladesh, **4** Médecins Sans Frontières, Operational Centre Brussels, Brussels, Belgium

* temmy.sunyoto@brussels.msf.org

## Abstract

Scabies is a dermatological parasitic infestation prevalent in many regions worldwide. Classified as a neglected disease by World Health Organization (WHO) since 2017, it is often associated with poor living conditions and overcrowding. Towards the end of 2021, unusual high numbers of scabies cases in outpatient consultations were observed in two Médecins Sans Frontières' (MSF) Primary Healthcare Centers (PHCs) in Rohingya refugee camps in Cox's Bazar, Bangladesh. Here, we aimed to describe the epidemiological and clinical characteristics of patients with scabies consulting the clinics from July 2022–November 2023. A cross-sectional study using routinely collected data from scabies' consultations at two MSF clinics located in camp 14 and 15 (total population 91,241 in 2023) was conducted. We retrospectively analyzed programmatic data of patients of all ages attending outpatient consultations and clinically diagnosed as scabies. Data were extracted from MSF clinical routine monitoring databases and descriptive statistics were reported. During the 16-month period, a total of 178,922 scabies consultations were recorded, amongst whom 57.7% were women and 42.3% men. Children <5 years constituted 20.5% of the cases, age-groups 6-14, 36.6%, and ≥15 years, 42.9%. Camp 15 had the highest number of cases (39.4%), followed by other camps (29.7%), and then camp 14 (24.4%). Most cases were simple scabies (79.5%); one in five were scabies with secondary infection cases. Patients were mainly treated with oral ivermectin (71.2%) and topical permethrin (24.3%); 19.5% of patients also received antibiotics. Our findings indicate that scabies is a significant health concern in the Cox's Bazar refugee camp. This study recorded over 178000 cases in the above period. The scale of this outbreak warrants further actions, including a prevalence survey, quality implementation of mass drug

**Data availability statement:** Data are available on request in accordance with MSFs data sharing policy. Requests for access to data should be made to data.sharing@msf.org. For more information please see: 1) MSF's Data Sharing Policy: https://www.msf.org/sites/msf.org/files/msf_data_sharing_policycontact_info-annexes_final.pdf2) MSF's Data Sharing Policy PLOS Medicine article: https://journals.plos.org/plosmedicine/article?id=10.1371/journal.pmed.1001562. Data is available and can be provided upon request to the corresponding author.

**Funding:** Médecins Sans Frontières (MSF) provided support in the form of salaries for MBAA, MSI, AD, SMS, MMH, KUM, MKM, TGP, KN, PP, SSN, YW, MM, LdS, MvH, LR, AB, TS. MSF programmatic funding covered all costs associated with the study. MSF was involved in the study design, data collection and analysis, decision to publish, and preparation of the manuscript.

**Competing interests:** The authors have declared that no competing interests exist.

administration, and multidisciplinary interventions related to camps' living conditions such as water and sanitation.

## Introduction

Scabies, a contagious skin infestation, is caused by a microscopic mite, *Sarcoptes scabiei* var. *hominis* that burrows into the upper layer of the skin. It is commonly characterized by intense itching, particularly at night, and a rash consisting of small red bumps (erythematous papules), scratch marks/excoriations, nodules, crusting, and/or vesicles. It is a common problem worldwide and can affect people of all ages, races, and socioeconomic classes [1–3]. Scabies is transmitted directly by skin-to-skin contact. The mite may also be transmitted indirectly through contact with contaminated bedding, clothing, or furniture [4]. Scabies can lead to secondary bacterial infections such as impetigo or bacteremia, mostly caused by *Streptococcus pyogenes* (group A *streptococcus*, GAS) and *Staphylococcus aureus*. Scabies may also lead to a range of downstream clinical complications, including invasive GAS infections, toxin-mediated diseases (e.g., scarlet fever, toxic shock syndrome), or autoimmune complications (e.g., rheumatic fever, glomerulonephritis) [5–8], emphasizing the need for prompt treatment to mitigate long-term health sequelae. Its diagnosis is primarily clinical by examination for pathognomonic features [9,10].

In 2017, scabies was recognized as a Neglected Tropical Disease (NTD) by the World Health Organization (WHO). Scabies occurs worldwide but is particularly problematic in areas of poor sanitation, overcrowding, and social disruption, with 200 million people estimated to suffer from scabies at any single time. Prevalence of scabies ranged from 0.2% to 71.4%, and is especially high in children, high in the Pacific and Latin American regions, with populations experiencing rates greater than 10% in all regions except for Europe and the Middle East [5,11–14]. Outbreaks of scabies are mostly reported from overcrowded settings such as refugee camps, prisons, schools, care/nursing homes, where close contact and shared facilities contribute to its spread [15]. In these settings, untreated scabies leads to severe complications, increasing morbidity and burdening already limited healthcare [16,17]. The intense itching and skin damage caused by scabies significantly reduce quality of life, adding to the daily suffering of affected individuals [18]. Addressing scabies is thus a critical public health need and a human rights imperative, as these vulnerable populations deserve basic healthcare and protection from preventable disease [11].

This burden of scabies in overcrowded settings, particularly in regions of poverty and displacement, can be highlighted by the plight of populations such as the Rohingya refugees, considered a stateless group, forced to move to Bangladesh after 2017 eruption of violence in Myanmar's Rakhine State [19]. There are currently 1.1 million Rohingya refugees in Bangladesh, accommodated in 34 congested camps, that are completely dependent on life-saving aid and services in Cox's Bazar district of Bangladesh [20]. The need for targeted health interventions in these settings is clear, but knowledge gaps around the size and scope of cases and what effective interventions can be implemented in such settings remain [11,21].

Médecins Sans Frontières (MSF) has established comprehensive programs to address population heath needs in Cox's Bazar refugee camps since the onset of the forced displacement in 2017-2018. In 2021, two primary health centers (PHC) of the Jamtoli project observed a gradual rise in skin infection cases, notably scabies, in outpatient department (OPD) consultations. Treated patients also reported a rise in the number of individuals with similar skin lesions in the community. In early 2022, this high burden was confirmed in a pilot survey conducted by MSF teams in which 28% of OPD consultations were skin

diseases; 83% of these skin diseases were identified as scabies based on clinical manifestation by trained medical assistants [22]. In response to this increase and to prevent an overload of cases hampering with other medical service delivery, MSF added a dedicated scabies clinic each in its 2 PHCs with a screen-and-treat approach accompanied by community-based health promotion activities consisting, three components of outreach activities – mental health, sexual and reproductive health, and noncommunicable diseases – that have been integrated together (outreach strategy). The catchment population of these health facilities were refugees living in camps 14, 15 and 16, totaling approximately 91,241 people (in 2023). The two MSF clinics observed an average of 300-600 cases a day with peaks recorded at 800 consultations per day.

This study aims to provide a description of the epidemiological and clinical characteristics of patients with scabies treated in the MSF scabies clinics in Rohingya refugee camps, Cox's Bazar, Bangladesh from July 2022 to November 2023.

## Materials and methods

### Study design

The study is a descriptive, cross-sectional study using facility-based data from an MSF project in Jamtoli, Cox's Bazar district, Bangladesh.

### Study setting and management of scabies

Since 2018, the MSF has supported two PHCs located in camp Hakimpara (camp 14) and Jamtoli (camp 15). These are two comprehensive first line health facilities offering services that include: Outpatient Department (OPD)/Emergency Room (ER), isolation, observation, vaccination, reproductive health, mental health, non-communicable diseases (NCD), including Hepatitis C. Outreach activities are also organized by Health Promotion team. The catchment population 91,241 *from camps 14, 15 and the adjacent camp 16 as of 2023 (Fig 1). Conditions in these camps are cramped with considerably limited water and sanitation facilities. The density of the population does not permit privacy while lack of maintenance renders many latrines and washing points dysfunctional. MSF is one of the many health actors and works in collaboration with relevant stakeholders and authorities.

In the two PHCs, after a sharp increase in scabies cases was noted, MSF envisioned the need for a separate service to ensure smooth and quality patient management. Patients who came to the PHCs for OPD consultation were triaged in a dedicated area; a crowd controller directed patients with skin complaints to scabies consultation booths in a separate wing of the PHCs. Trained medical assistants diagnosed patients clinically using standard criteria [9]. Patients were asked to bring their household members, even if asymptomatic, to the clinic to receive treatment. Medication was dispensed in dedicated areas by the drug dispensers; information on the dispensed drug was encoded in the consultation card by a dedicated person. Locally adapted hygiene messages were given in the waiting area by community health workers.

The medical protocol used for treating scabies was 200 µg/kg of body weight of oral ivermectin, with a second dose given 7-14 days later, and this is the same for contacts (members of the same household). For those contraindicated for ivermectin (e.g., children weighing less than 15 kg, pregnant women), topical permethrin (5% cream) was given, with instructions to apply all over the affected skin area and to leave it in place for 8-12 hours [23]. The first dose of ivermectin was given with the Direct Observation Therapy approach; the second dose was given in a small envelope to be taken at home. Health education messages on prevention of transmission were given by community health volunteers.

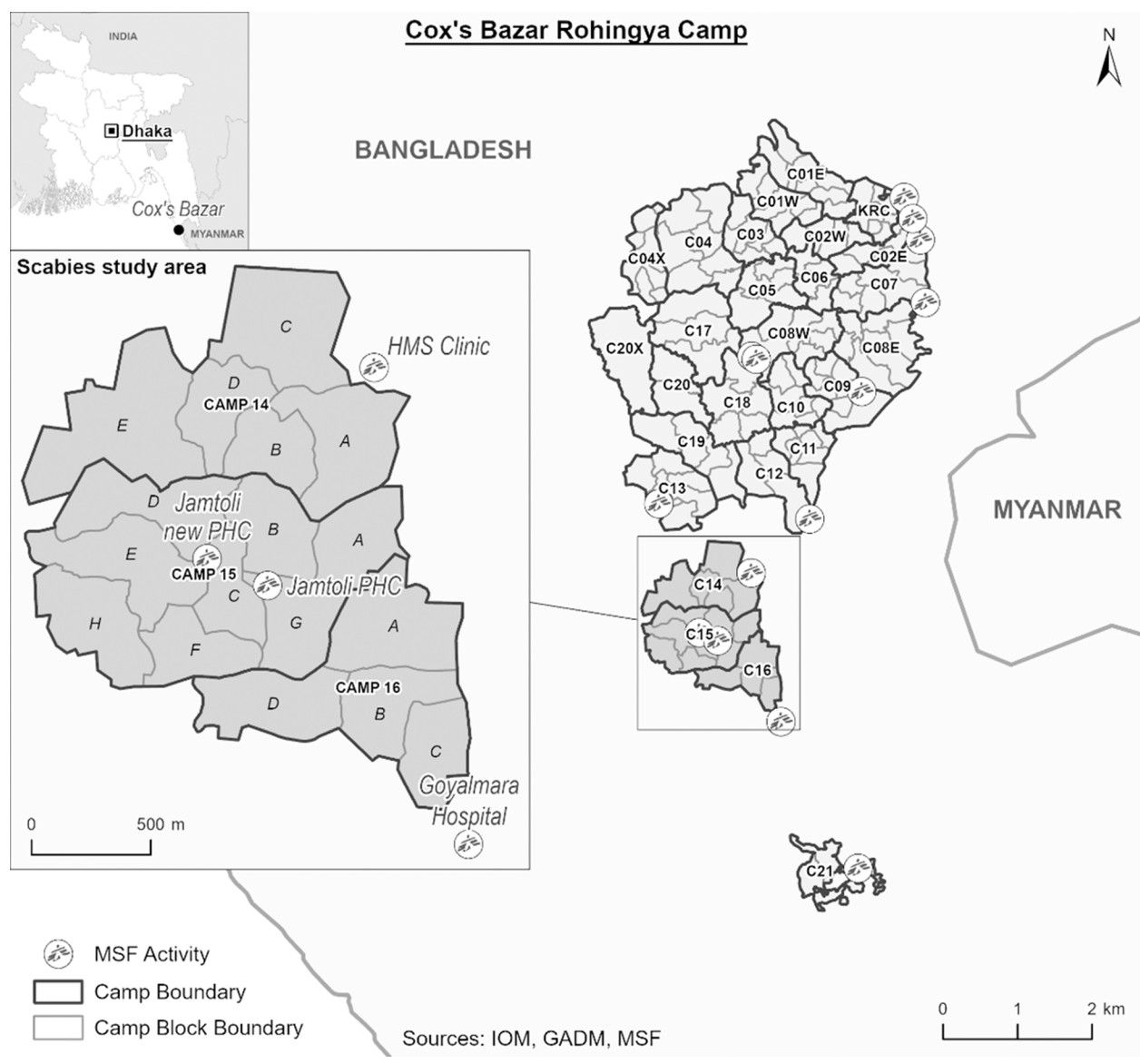

**Fig 1. Map of Cox's Bazaar Rohingya Refugee Camps with MSF facilities location.**

## Study participants

All individuals of all ages who sought care at the scabies clinic of the PHC center of MSF Jamtoli project, Rohingya Refugee Camp, Cox's Bazar, Bangladesh from 1st July 2022 to 15th November 2023 were included.

## Data management and analysis

Programmatic data is collected routinely and used to monitor trends and assess the services quality to ensure sufficient resources. The OPD consultation data was entered directly by dedicated staff at the scabies clinic, supervised by the data manager. Data was collected with the Kobo Toolbox extracted, cleaned using MS Excel, and analyzed using SPSS statistical software

V22. The research team were given access to the de-identified data during the months of December 2023 and January 2024.

Variables included in the analyses consisted of demographic information (age in years, sex, camp origin) and clinical information (new cases or reinfestation or contact first time seen in MSF clinic) and treatment received. Reinfestation was defined as if the symptoms disappeared after receiving treatment, but similar symptoms start appearing again after four weeks (1 month). Simple cases were those with no secondary bacterial infection, while sur-infected scabies is termed 'impetiginized scabies'. Descriptive results are expressed in absolute numbers and percentages for categorical variables, and median and interquartile rage (IQR) for numerical variables.

### Ethical Approval statement:

This research fulfilled the exemption criteria set by the Médecins Sans Frontières Ethics Review Board for a posteriori analysis of routinely collected clinical data (ERB2346). It was conducted with permission from Medical Director, Operational Center Brussels and received ethical clearance from ERB University of Creative Technology Chittagong, Bangladesh.

### Results

The total number of scabies consultations registered between July 2022 and November 2023 were 178,922. A peak in the number of consultations was noted in epidemiological week 35-2022 (3922 cases), and week 4-2023 (3334 cases), followed by slight decline in week 26-2023 (1069 cases), before it increased again to 3137 cases in week 38-2023. The highest daily peak in consultations was observed in week 3, 2023 at 640 cases per day, number of cases typically ranged between 300-600 cases per day. Fig 2 illustrates the monthly distribution of scabies consultations from July 2022 to November 2023.

Table 1 represents the demographic distribution of scabies consultations. Patients who were women comprised most cases at 57.7%, while patients who were men constituted 42.3%. The 6-14 age-group was the most affected at 36.6%, followed closely by those over 15 at 42.9%, and the 0-5 age-group at 20.5%. The majority of individuals were Rohingya refugees, accounting for 93.5% of cases, with Bangladeshi nationals representing a minority at 6.5%.

Cases that came from Camp 15 made up a total of 39.4% from all consultations, with Camp 14 and 'Other' camps contributing 24.4% and 29.7% respectively, and Camp 16 at 6.5%. New scabies cases formed most consultations at 72.2%, with re-infections and contacts constituting 20.1% and 7.7%, respectively. The case type predominantly observed was simple scabies at 79.5%, with the remaining 20.5% classified as scabies with secondary infection/ impetiginized scabies.

Regarding the treatment modalities for scabies within the cohort, the data from Table 1 reveals that 1 in 4 patients (24.3%) received Permethrin. Conversely, a substantial majority of 71.2% received Ivermectin as part of their treatment regimen. Antibiotics were administered less frequently, with 19.5% of the cases receiving such treatment. Similar findings were observed with the use of antihistamines, wherein 81% of the patients did not receive this class of medication.

Table 2 represents the results of the logistic regression analysis between the research variables (gender, age group, case definition, and use of antibiotic) of patients with simple or impetiginized scabies. The age group and case definition variables were categorized into (over 5 year, 5 years or younger) and (new case, reinfection/contact) retrospectively. Patient gender did not show a significant difference between simple or complicated cases ($p>0.05$); however, men had slightly higher odds of developing complicated scabies (OR= 0.75, CI=0.74–0.77).

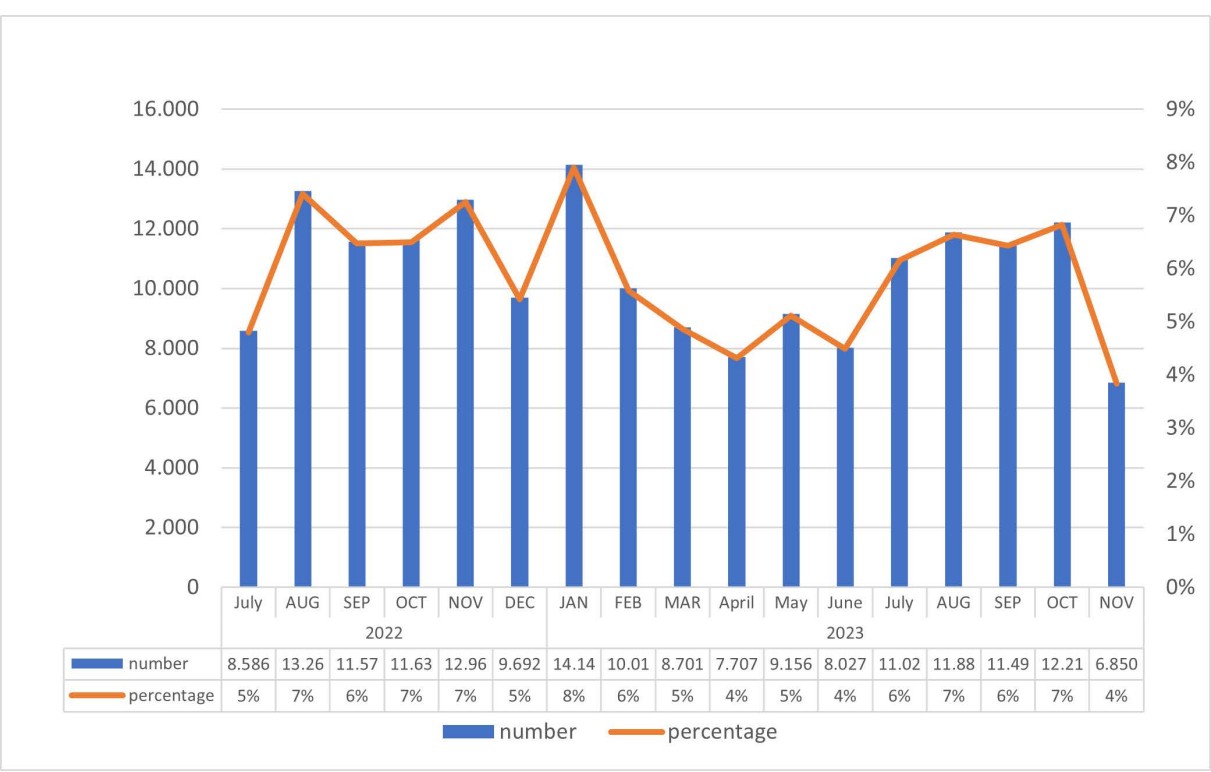

**Fig 2. Number of scabies consultations per month in MSF PHCs between July 2022-November 2023 (total n =178,922).**

Table 1 . Number (%) of Scabies consultation per camp/gender/ age group.

| Variables | | | Count (%) |
|---|---|---|---|
| **Sex** | | female | 103,415 (57.7%) |
| | | male | 75,507 (42.3) |
| **Age groups** | | 0-5 | 36,634 (20.5%) |
| | | 6-14 | 65,454 (36.6%) |
| | | over 15 | 76,834 (42.9%) |
| **Nationality** | | **Bangladeshi** | 11,654 (6.5%) |
| | | **FDMN** | 167,268 (93.5%) |
| **Camp** | | **Other** | 53,153 (29.7%) |
| | | **Camp 14** | 43,619 (24.4%) |
| | | **Camp 15** | 70,496 (39.4%) |
| | | **Camp 16** | 11,654 (6.5%) |
| **Case selection** | | new case | 129,128 (72.2%) |
| | | re-infection | 35,931 (20.1%) |
| | | contact | 13,863 (7.7%) |
| **Case type** | | simple scabies | 141,610 (79.5%) |
| | | Secondary infected/impetiginized | 36,552 (20.5%) |
| **Treatment** | **Permethrin** | Yes | 43,441 (24.3%) |
| | **Ivermectin** | Yes | 127,320 (71.2%) |
| | **Antibiotics** | Yes | 34,874 (19.5%) |
| | **Antihistaminic** | Yes | 33,958 (19%) |

**Table 2. Analysis of risk factors and complications in scabies cases.**

| | | simple scabies | complicated | Regression model | |
|---|---|---|---|---|---|
| | | Count (%) [+] | Count (%) | Odd ratio (95% CI)[1] | P-value |
| Sex | Female | 84,352 (81.9%) | 18,600 (18.1%) | 1 | 0.912 |
| | Male | 57,258 (76.1%) | 17,952 (23.9%) | 0.75 (0.74 – 0.77) | |
| Age group | Over 5 years | 103,357 (82.3%) | 22,191 (17.7%) | **1** | **0.012 *** |
| | 5 years or younger | 25,406 (69.5%) | 11,162 (30.5%) | **1.72 (1.69 – 1.76)** | |
| Cases | Reinfection or contact | 40,594 (82.8%) | 8,440 (17.2%) | **1** | **0.043 *** |
| | New cases | 101,016 (78.2%) | 28,112 (21.8%) | **1.26 (1.23 – 1.29)** | |
| Antibiotic | No | 140,188 (97.8%) | 3,114 (2.2%) | **1** | **0.000 *** |
| | Yes | 1,422 (4.1%) | 33,438 (95.9%) | **23.9 (22.7- 25.5)** | |

[1]Adjusted for impetiginized/scabies cases with secondary infection.

[+] Raw percentage (within the same variable)

* Significant (P<0.05)

On the other hand, age group showed significant difference (p=0.012); patients under 5 years of age showed a higher risk of developing impetiginized scabies (OR=1.72, CI= 1.69–1.76). Similarly, new cases had higher risk of being secondarily infected cases compared to the reinfection of contact (P=0.043), (OR=1.26, CI= 1.23–1.29). The use of antibiotics was mainly associated with the impetiginized cases of scabies (95.9%). Patients with impetiginized scabies were significantly more likely to use antibiotics (P=0.0001), (OR=23.9, CI= 22.7-25.5).

## Discussion

This study describes the unusually high numbers of consultations and cases observed in MSF clinics in camps 14 and 15 in Cox's Bazar, Bangladesh from July 2022 to November 2023.. Scabies, despite being a highly contagious skin infection, is usually easily treatable. However, the context within the Rohingya refugee camps (e.g., crowded living conditions, limited water and sanitation facilities, and barriers to quality medical care) make the situation optimal for outbreaks to occur and widespread transmission throughout the camp. Harsh living conditions and limited resources make it challenging to effectively manage and control an outbreak, making scabies increasingly concerning for refugees living in Cox's Bazar [20].

In a period of 18 months, close to 179,000 consultations were conducted, with one fifth of the total number of cases presenting complication, necessitating a specific service diagnose and treat scabies cases among the residents of the camp and the local community [22] to be prevent an overload within the MSF clinics. The results of analyzed of routine data collected from MSF scabies clinics showed a high number of scabies cases since November 2021. The number of the cases peaked in March 2022 with over 14,000 cases. The total number of scabies consultations curbed during March 2022 and March 2023; however, the number of cases remained exceptionally high with over 10,000 consultations per month except for April, May and December 2022. Previous literature has documented scabies outbreaks in closed context such as islands and institutions [24–26] but the scale as reported in this study is unprecedented in refugee camps. Nevertheless, the clinical characteristics of scabies in patients attending consultations in the Cox's Bazar refugee camps align with the disproportionate impact of scabies on disadvantaged populations, as observed in other settings [16,27,28].

Despite awareness and efforts by health actors in the camps, the rate of scabies consultations at MSF clinics remains high. This could be attributed to the inconsistent access to scabies treatments and continuous spread between camps. Apart from case management for scabies

at health facilities, MSF also conducted environmental decontamination (washing the clothes and bedding, followed by spraying with chlorine solution) in selected households and madrasas (education centers) in these two camps. Nevertheless, these interventions did not appear to have any significant impact on the total number of the cases despite the success in spreading health awareness and hygiene kits [29].

The treatment of scabies cases with oral ivermectin in our study is consistent with other studies on the efficacy of ivermectin in controlling scabies in other populations, such as asylum seekers that reside in confinements [17,30,31]. Permethrin, on the other hand, might present challenges in environments with limited water access and reliance on soap distribution such as in refugee camps, and a reality in Cox's Bazar. Although its application does not strictly require bathing beforehand, the recommendation to apply permethrin on all skin surfaces and remove it after 8–14 hours through washing may pose significant difficulties without adequate water resources [32]. In water-scarce settings, thorough cleansing post-treatment might not be feasible, which could reduce treatment efficacy, increase the risk of residual irritation, or complicate hygiene practices.

Antibiotic was used as additional treatment for the impetiginized cases which presented scabies-related bacterial superinfection and is usually caused by *Staphylococcus aureus* and *Streptococcus pyogenes* (group A streptococcus) [33]. In cases of scabies complicated by bacterial infections, standard treatment for impetigo includes topical application of mupirocin for localized infections and oral antibiotics, such as cefalexin or cloxacillin, for more extensive or complicated cases. In neonates with lesions around the umbilicus, intravenous cloxacillin is administered to mitigate the risk of systemic infection [23]. Although these regimens aim to effectively manage scabies-related bacterial complications, we unfortunately lack follow-up data to confirm treatment outcomes for this cohort.

Mass Drug Administration (MDA) has been suggested as a successful intervention against scabies in high prevalence settings where individual detection and treatment of scabies is not considered feasible on a large scale. While the use of MDA for the control of diseases is not new, this approach gained prominence for some NTDs in the 1990s. Diseases such as onchocerciasis, lymphatic filariasis, trachoma, schistosomiasis, and soil-transmitted helminths are amenable to MDA because of the availability of safe and affordable drugs. MDA involves the distribution of medications to entire populations in the respective community, including those who are not infected with the aims to prevent transmission and reduce the disease burden.

WHO currently recommends MDA for scabies in populations where prevalence is >10% [34,35],. While ivermectin is the drug of choice for scabies during an MDA, there are contraindications for its use, and use of alternative such as permethrin as topical treatment poses significant challenges. Application requires treating all skin areas, including the scalp and under the nails, ideally on cool, dry skin and left for 8–14 hours. Limited access to private spaces, clean clothing, and resources to ensure all close contacts receive simultaneous treatment complicates adherence, increasing the risk of reinfestation and reducing treatment efficacy.

Following the original landmark studies of scabies community control in Panama in the 1970s and 1980 using lindane followed by permethrin, ivermectin was used in place of permethrin (1997), except in children weighing <15 kg and pregnant women. All children were re-examined and treated at 6-monthly intervals. The prevalence of scabies reduced from 25% to < 1% over 3 years, with a concomitant reduction in impetigo [36,37]. Fiji's 2012 Skin Health Intervention Fiji Trial (SHIFT) compared MDA with ivermectin, permethrin, or standard care: 12 months post-MDA, scabies prevalence decreased by 94% with ivermectin, 62% with permethrin, and 49% with standard care [38]. A survey 24-month post-MDA showed sustained low scabies prevalence (3.6%) with ivermectin [39]. The Solomon Islands'

AIM trial used ivermectin-azithromycin, reducing scabies prevalence by 88% at 12 months, remaining 74.9% lower than baseline at 36 months. Evidence suggested similar effect sizes for ivermectin-based and permethrin MDA [40]. These findings underscore the efficacy of MDA approaches, with ivermectin demonstrating notable effectiveness in scabies control trials, while 5% permethrin remains more costly and complex to distribute logistically. A 2020 systematic review of MDA reports that MDA for scabies, across a diverse range of study designs and settings, led to a relative reduction of 79% in scabies and 66% in impetigo following MDA. In populations with a prevalence of scabies>10%, there was a greater reduction in scabies indicating that MDA may be more effective in this population [41]. In Ethiopia, the largest MDA campaign was conducted (with ivermectin) in 2018 for an outbreak associated with drought – they demonstrated that community engagement was imperative [42,43]. Despite this, there remain gaps in terms of evidence around the implementation of different intervention toolboxes in specific context, making it difficult to influence policy changes on a wider scale and offer specific longer term and implementable solutions for (large) refugee settlements such as the Rohingya refugee camps in Cox's Bazar and other. In June 2022, a WHO rapid assessment in Cox's Bazar reported that the scabies prevalence did not yet meet the 10% threshold for MDA [29], while a 2023 study claimed the prevalence of 67% [44]. Between December 2023 and February 2024, an MDA campaign was finally implemented in Cox's Bazar, that brought prevalence as well as consultation significantly down [45]. However, the long-term effects are unknown, and cases will likely rise again over time if no additional measures are implemented. Effective MDA requires strong community engagement; in Cox's Bazar, health promotion efforts and risk communication were tailored to the specific context, with the involvement of community health workers and camp leaders proving indispensable.

Scabies outbreaks, far from being benign, are associated with mortality and impose significant health burdens—including severe physical and mental distress, pyoderma, chronic renal disease leading to lifelong hypertension, and an increased risk of heart disease. Although scabies is treatable and preventable, resource limitation, cross-contamination, implementation constraints and a lack of coordinated responses contribute to high case loads and epidemic levels of infestation. Further operational research is required to assess the effectiveness and methods of implementation of suitable intervention options such as combined MDA approaches and environmental (WASH) interventions, as well as effectively monitoring trends in humanitarian settings such as the refugee camps in Cox's Bazar. This study underscores the challenges of managing scabies outbreaks in camps where inadequate WASH infrastructure, limited access to clean water, and poor sanitation persist. Strengthening WASH interventions, including the provision of functional washing facilities, soap, and well-maintained latrines, is crucial. Coupled with appropriate community health education, these improvements could significantly enhance scabies control efforts in these vulnerable settings. Effective control requires coordinated management of contacts and ensures that both refugees and local resident populations benefit from intervention efforts.

This study is a retrospective analysis of routinely collected data during an outbreak that represented an extremely high number of consultations at the MSF health facilities in Cox's Bazar – therefore several limitations were acknowledged. Due to the high caseload, it was not possible to collect detailed data such as patients' behavior, household level infections, transmission dynamics and treatment outcomes. Some variables that may have been pertinent to be included in the analyses could not be collected. Additionally, this facility-based study only included scabies patients who sought care at MSF-OCB clinics and are not necessarily represent the broader population of the camps suffering from scabies. Individuals who did not seek treatment, whether due to access barriers or mild symptoms, were not captured. Moreover, this study used retrospective data already collected and encoded in project databases,

while certain variables, such as socioeconomic factors or environmental conditions as potential confounders, were absent and could not be included in the analysis.Lastly, the inability to assess scabies prevalence at the community level during the study period limits the capacity to understand the full scope of the outbreak and the potential impact of interventions.

## Conclusion

The scabies outbreak in Cox's Bazar refugee camp is a significant health concern for the over one million Rohingya refugees living in extremely difficult conditions. Despite ongoing efforts, high case loads persist, exposing gaps in coordinated management. Immediate actions—comprehensive surveys, coordinated contact management, and innovative interventions combining MDA, WASH improvements, and community education—are essential to curb transmission and prevent long-term complications in both displaced and local populations.

## Acknowledgements

We would like to thank the GIS Team for providing us with a map (Fig 1). Additionally, we would like to acknowledge the MSF team and its community members that have worked hard to address the scabies problem with dedications.

## Author contributions

**Conceptualization:** MHD Bahaa Aldin Alhaffar, Asish Das, Trine Grondahl Petersen, Pankaj Paul, Srijeeth S Nair, Yves Wailly, Marta Miazek, Letizia di Stefano, Michel van Herp, Temmy Sunyoto.

**Data curation:** MHD Bahaa Aldin Alhaffar, Md. Soriful Islam, Shakil Mohammed Shibloo.

**Formal analysis:** MHD Bahaa Aldin Alhaffar, Md. Soriful Islam, Shakil Mohammed Shibloo.

**Funding acquisition:** Asish Das, Trine Grondahl Petersen, Srijeeth S Nair, Yves Wailly, Marta Miazek, Temmy Sunyoto.

**Investigation:** Md. Soriful Islam, Asish Das, Shakil Mohammed Shibloo, Md. Mahmudul Hasan, Kamar Uddin Muzakkir, Md. Kawser Mahmud, Trine Grondahl Petersen, Pankaj Paul, Srijeeth S Nair, Temmy Sunyoto.

**Methodology:** Mohammad Injamul Hoq, Asish Das, Shakil Mohammed Shibloo, Pankaj Paul, Letizia di Stefano, Michel van Herp, Amrish Baidjoe.

**Project administration:** Mohammad Injamul Hoq, Asish Das, Md. Mahmudul Hasan, Kamar Uddin Muzakkir, Md. Kawser Mahmud, Trine Grondahl Petersen, Karsten Noko, Pankaj Paul, Srijeeth S Nair, Yves Wailly, Marta Miazek, Amrish Baidjoe.

**Resources:** Asish Das, Md. Mahmudul Hasan, Kamar Uddin Muzakkir, Trine Grondahl Petersen, Karsten Noko, Pankaj Paul, Yves Wailly, Marta Miazek, Amrish Baidjoe, Temmy Sunyoto.

**Software:** Mohammad Injamul Hoq.

**Supervision:** Mohammad Injamul Hoq, Trine Grondahl Petersen, Karsten Noko, Pankaj Paul, Srijeeth S Nair, Yves Wailly, Marta Miazek, Letizia di Stefano, Amrish Baidjoe, Temmy Sunyoto.

**Validation:** Mohammad Injamul Hoq, Karsten Noko, Pankaj Paul, Srijeeth S Nair, Yves Wailly, Marta Miazek, Letizia di Stefano, Michel van Herp, Temmy Sunyoto.

**Visualization:** MHD Bahaa Aldin Alhaffar, Letizia di Stefano, Lekha Rathod, Amrish Baidjoe.

**Writing – original draft:** MHD Bahaa Aldin Alhaffar.

**Writing – review & editing:** Md. Soriful Islam, Mohammad Injamul Hoq, Shakil Mohammed Shibloo, Pankaj Paul, Yves Wailly, Marta Miazek, Letizia di Stefano, Michel van Herp, Lekha Rathod, Amrish Baidjoe, Temmy Sunyoto.

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
