## [Decision Letter · Decision Letter 0]

15 Oct 2024

PGPH-D-24-01980

High caseload of Scabies amongst Rohingya refugees in Cox’s Bazar, Bangladesh: A retrospective analysis of the epidemiological and clinical characteristics of cases, July 2022 to November 2023

Dear Dr. Sunyoto,

Thank you for submitting your manuscript to PLOS Global Public Health. After careful consideration, we feel that it has merit but does not fully meet PLOS Global Public Health’s publication criteria as it currently stands. Therefore, we invite you to submit a revised version of the manuscript that addresses the points raised during the review process.

We look forward to receiving your revised manuscript.

Kind regards,

Andrés F. Henao-Martínez, M.D.

Academic Editor

Journal Requirements:

1. Some material included in your submission may be copyrighted. According to PLOS’s copyright policy, authors who use figures or other material (e.g., graphics, clipart, maps) from another author or copyright holder must demonstrate or obtain permission to publish this material under the Creative Commons Attribution 4.0 International (CC BY 4.0) License used by PLOS journals. Please closely review the details of PLOS’s copyright requirements here: PLOS Licenses and Copyright. If you need to request permissions from a copyright holder, you may use PLOS's Copyright Content Permission form.

Potential Copyright Issues:

Figure 1: please (a) provide a direct link to the base layer of the map (i.e., the country or region border shape) and ensure this is also included in the figure legend; and (b) provide a link to the terms of use / license information for the base layer image or shapefile. We cannot publish proprietary or copyrighted maps (e.g. Google Maps, Mapquest) and the terms of use for your map base layer must be compatible with our CC-BY 4.0 license. 

Additional Editor Comments (if provided):

Reviewers' comments:

Reviewer's Responses to Questions

**Comments to the Author**

1. Does this manuscript meet PLOS Global Public Health’s publication criteria ? Is the manuscript technically sound, and do the data support the conclusions? The manuscript must describe methodologically and ethically rigorous research with conclusions that are appropriately drawn based on the data presented.

Reviewer #1: Yes

Reviewer #2: Yes

2. Has the statistical analysis been performed appropriately and rigorously?

Reviewer #1: Yes

Reviewer #2: Yes

3. Have the authors made all data underlying the findings in their manuscript fully available (please refer to the Data Availability Statement at the start of the manuscript PDF file)?

Reviewer #1: No

Reviewer #2: Yes

4. Is the manuscript presented in an intelligible fashion and written in standard English?

Reviewer #1: Yes

Reviewer #2: Yes

5. Review Comments to the Author

Reviewer #1: Overall, the manuscript is well-written and organized, but there are a few areas where clarification and additional discussion could strengthen the paper.

The introduction provides and adequate background but please provide more in-depth context about the importance of addressing scabies in refugee settings, emphasizing the public health and human rights aspects of this issue.

The manuscript provides detailed information about the treatment regimens used (i.e., ivermectin and permethrin). However, more discussion about the limitations of these treatments, particularly in the context of limited water access for permethrin application, would be valuable. The paper also briefly mentions antibiotic use for complicated cases, but further detail on the types of bacterial infections treated and the outcomes for those patients would enrich the findings.

Expand the discussion on the challenges and future directions for interventions in similar refugee settings. Specifically, consider elaborating more on the WASH interventions and any gaps in community health education that might need to be addressed.

Address the limitations in a more structured way, including any biases or confounding factors that may have impacted the study results.

Reviewer #2: Some limited concerns

1. Cox Bazar vs Cox's Bazar – please review for consistency throughout

2. Discordance of stated population numbers – three different numbers are mentioned in relation to the population (as well as only two camps vs more than two camps). This does get confusing at times, please clarify. Specifically, the abstract, the methods, and the results differ in number.

6. PLOS authors have the option to publish the peer review history of their article (what does this mean? ). If published, this will include your full peer review and any attached files.

**Do you want your identity to be public for this peer review?** For information about this choice, including consent withdrawal, please see our Privacy Policy .

Reviewer #1: No

Reviewer #2: No

---

## [Decision Letter · Decision Letter 1]

26 Feb 2025

PGPH-D-24-01980R1

High caseload of Scabies amongst Rohingya refugees in Cox’s Bazar, Bangladesh: A retrospective analysis of the epidemiological and clinical characteristics of cases, July 2022 to November 2023

Dear Dr. Sunyoto,

Thank you for submitting your manuscript to PLOS Global Public Health. After careful consideration, we feel that it has merit but does not fully meet PLOS Global Public Health’s publication criteria as it currently stands. Therefore, we invite you to submit a revised version of the manuscript that addresses the points raised during the review process.

The reviewer has provided some valuable comments for you to review and address, they specifically mention improving the clarity around the message you want to put across with this research. The comments can be found

below, please address their concerns in your revised manuscript. 

We look forward to receiving your revised manuscript.

Kind regards,

Emma Campbell, Ph.D

Staff Editor

Additional Editor Comments (if provided):

Reviewers' comments:

Reviewer's Responses to Questions

**Comments to the Author**

1. If the authors have adequately addressed your comments raised in a previous round of review and you feel that this manuscript is now acceptable for publication, you may indicate that here to bypass the “Comments to the Author” section, enter your conflict of interest statement in the “Confidential to Editor” section, and submit your "Accept" recommendation.

Reviewer #3: All comments have been addressed

2. Does this manuscript meet PLOS Global Public Health’s publication criteria ? Is the manuscript technically sound, and do the data support the conclusions? The manuscript must describe methodologically and ethically rigorous research with conclusions that are appropriately drawn based on the data presented.

Reviewer #3: Yes

3. Has the statistical analysis been performed appropriately and rigorously?

Reviewer #3: Yes

4. Have the authors made all data underlying the findings in their manuscript fully available (please refer to the Data Availability Statement at the start of the manuscript PDF file)?

Reviewer #3: No

5. Is the manuscript presented in an intelligible fashion and written in standard English?

Reviewer #3: Yes

6. Review Comments to the Author

Reviewer #3: Thanks for the opportunity to review this revised manuscript. I hope my comments are useful.

I think being very clear about what messages you want to get across with this analysis is important. They could be made more strongly.

eg.

Scabies is a huge problem in displaced persons camps particularly in the young.

Cross contamination and lack of co-ordinated response contribute to ongoing high case loads and epidemic levels of infestation.

Co-ordinated management of contacts is crucial to gain control.

Not managing scabies outbreaks has considerable potential to long term adverse health outcomes and even mortality

The local resident population also needs to be considered and need to benefit from services being provided for displaced persons (you mention this but do not emphasise this is an important consideration)

You many want to bring out other aspects

Following are some specific points:

Abstract:

Small point but you indicated scabies is classified as a neglected disease. By whom and where is the reference. I think you mean classified by WHO as a neglected tropical disease…..

What is “complicated/sur-infected” do you mean super-infected or impetiginized? If so suggest simplifying and just stating this.

Introduction:

It is not that rare for the downstream complications of scabies such as glomerulonephritis and I think it is the key message for emphasizing the importance of addressing scabies in the risk to long term health sequele.

For instance, symptomatic acute glomerulonephritis was reported in 10% of children in a survey in northern Australia, but, in addition, 24% had microscopic haematuria L Streeton, JN Hanna, RD Messer An epidemic of acute post‐streptococcal glomerulonephritis among Aboriginal children

J Paediatr Child Health, 31 (2008), pp. 245-248

You introduce the term mass drug administration in your introduction and the indications for when this would be recommended but then express the results that meant you did not actually use it. So suggest moving this aspect/point to the discussion area. It is already there, I am not clear it required in the introduction.

Methods

What are your locally adapted health messages given by the community health volunteers? This might be useful to share ( even if included as an addendum)

The term “complicated case” just refers to bacterial infection is that correct? It is not quite clear – if this is the case I suggest you just say impetiginized scabies or secondarily infected scabies. You do indicate in the methods that “simple” means “not infected” - once you have clarified this it will need to be consistently expressed in the manuscript.

Management of scabies: you don’t indicate what protocol you used for contacts of the cases is it possible to state and do you have information on the number of contacts treated per case?

In the results section you indicate that 20% of those treated were contacts – this is low – ie suggesting that the majority of contacts were not treated which would probably explain why there was so little impact on the continuing high level of cases…..

Also I suspect there is significant population exchange between the various camps which will contribute to the ongoing spread of cases

Discussion

217: you indicate that such high outbreak levels have not been previously reported and yet you reference the Ethiopian outbreak where there were 379,000 confirmed cases of scabies was identified the Amhara Ethiopian outbreak Enbiale W, Ayalew A. Investigation of a Scabies Outbreak in Drought-Affected Areas in Ethiopia. Trop Med Infect Dis. 2018 Oct 29;3(4):114. doi: 10.3390/tropicalmed3040114. PMID: 30380650; PMCID: PMC6306922. Also the authors went onto administrer MDA

Enbiale W, Baynie TB, Ayalew A, Gebrehiwot T, Getanew T, Ayal A, Ayalew M, De Vries HJ, Takarinda K, Manzi M, Zachriah R. "Stopping the itch": mass drug administration for scabies outbreak control covered for over nine million people in Ethiopia. J Infect Dev Ctries. 2020 Jun 29;14(6.1):28S-35S. doi: 10.3855/jidc.11701. PMID: 32614793. This would be relevant to reference and discussion in your context.

224: what environmental decontamination measures were conducted? What is the evidence of benefit anyway for such an approach? Suggest you include

261 “Scabies outbreaks, although not attributing to mortality cause significant added burdens on 262 populations including physical and mental distress”

I disagree, scabies has an associated mortality – important not to underplay in such a setting.

Pyoderma, chronic renal disease leading to life long hypertension and increased heart disease…. See figure in Toward the Global Control of Human Scabies: Introducing the International Alliance for the Control of Scabies | PLOS Neglected Tropical Diseases (https://journals.plos.org/plosntds/article?id=10.1371/journal.pntd.0002167)

Which illustrates modelled mortality data for scabies.

290: scabies does have a mortality see above….so don’t underplay this.

I hope these comments are useful to enhance the key messages arising from this analysis.

7. PLOS authors have the option to publish the peer review history of their article (what does this mean? ). If published, this will include your full peer review and any attached files.

**Do you want your identity to be public for this peer review?** For information about this choice, including consent withdrawal, please see our Privacy Policy .

Reviewer #3: **Yes: ** Dr Lucinda Claire Fuller

---

## [Editor Report · Decision Letter 2]

7 Mar 2025

High caseload of Scabies amongst Rohingya refugees in Cox’s Bazar, Bangladesh: A retrospective analysis of the epidemiological and clinical characteristics of cases, July 2022 to November 2023

PGPH-D-24-01980R2

Dear Dr Sunyoto,

We are pleased to inform you that your manuscript 'High caseload of Scabies amongst Rohingya refugees in Cox’s Bazar, Bangladesh: A retrospective analysis of the epidemiological and clinical characteristics of cases, July 2022 to November 2023' has been provisionally accepted for publication in PLOS Global Public Health.

Best regards,

Andrés F. Henao-Martínez, M.D.

Academic Editor